# Predictors for the Development of Major Adverse Limb Events after Percutaneous Revascularization—Gender-Related Characteristics

**DOI:** 10.3390/medicina59030480

**Published:** 2023-02-28

**Authors:** Horatiu Comsa, Gabriel Gusetu, Gabriel Cismaru, Bogdan Caloian, Radu Rosu, Dumitru Zdrenghea, Adina David, Bogdan Dutu, Raluca Tomoaia, Florina Fringu, Diana Irimie, Dana Pop

**Affiliations:** 1Department of Internal Medicine, Cardiology Rehabilitation, Faculty of Medicine, University of Medicine and Pharmacy “Iuliu Hatieganu”, 400347 Cluj-Napoca, Romania; 2Cardiology Department, Clinical Rehabilitation Hospital, 400066 Cluj-Napoca, Romania; 3Ares Cardiomed, 400015 Cluj-Napoca, Romania

**Keywords:** peripheral artery disease, percutaneous angioplasty, major adverse limb events, women

## Abstract

*Background and Objectives*: Revascularization has been proven to be superior to medication for symptom improvement in patients with peripheral arterial disease (PAD). There are well known gender differences in therapeutic strategies for PAD. The influence of gender on post-angioplasty prognosis is not fully understood though. The present study aims to identify potential peculiarities between men and women undergoing peripheral angioplasty, as well as factors responsible for those differences. *Material and methods*: 104 consecutive subjects (50 women and 54 men) who underwent percutaneous angioplasty (PTA) between January and October 2019 for symptomatic PAD were included. Demographics, PAD history, cardiovascular risk factors, comorbidities, the associated coronary or cerebrovascular diseases, biological parameters, drug-treatment and PTA type and technique were taken into account. The follow-up period was 2 years, during which major adverse limb events (MALE) were documented. *Results*: The mean age was 67 ± 10 years. Women were 4 years older than the men (69 ± 10 years vs. 65 ± 9.2 years—*p* = 0.04). Smoking was more prevalent in men (*p* = 0.0004), while other cardiovascular risk factors did not differ significantly. The mean follow-up of the two groups was 21 ± 2.4 months. Women had infra-inguinal involvement more frequently (78%), while men exhibited mixed disease, with supra + infra-inguinal (37%) or solely supra-inguinal (20.3%) involvement (*p* = 0.0012). Rates of MALE were similar in the two groups (*p* = 0.914). Gender did not influence the incidence of PAD-related adverse events. The only parameter that proved to have a significant influence on the occurrence of MALE was the ankle–brachial index (ABI). A value below 0.5 was found to be an independent predictor for MALE (*p* = 0.001). *Conclusions*: There was no significant difference in the incidence rates of MALE between the two genders over a 2-year follow-up period post-PTA. Regardless of sex, an ankle–brachial index value below 0.5 was the sole independent predictor for limb-related adverse events.

## 1. Introduction

There are proven gender differences in terms of pathophysiology, clinical presentation and therapeutic options for peripheral arterial disease (PAD). Risk factors and epidemiological data related to PAD also seem to be influenced by sex. Women are generally older at first diagnosis and have a higher prevalence of diabetes mellitus; men, on the other hand, are more frequently active smokers and generally present with stable exertional symptoms [1,2]. Identifying and understanding these differences has attracted increasing interest among the scientific community. In recent decades, endovascular procedures have grown exponentially, as they are less invasive and require a far shorter recovery time than traditional open surgery. However, available data points towards lower usage of percutaneous procedures (PTA) among women with clinically manifest PAD, for several potential reasons. The most plausible ones are linked to anatomical peculiarities (smaller-caliber arteries) and clinical aspects (more frequently atypical symptoms resulting in incorrect or delayed diagnosis) [3]. The influence of gender on post-angioplasty prognosis is not fully understood. Previous studies have shown that women undergoing percutaneous coronary revascularization have higher rates of periprocedural complications and poorer long-term outcomes compared to men [4]. These differences can be attributed to older age at presentation, smaller vessels, the presence of comorbidities or higher anatomical complexity, all of which are observed to a greater extent in women. However, current data do not paint a clear picture when it comes to percutaneous procedures for PAD. Although there is a wealth of literature data showing gender-specific aspects of different revascularization strategies, few studies have directly compared outcomes by gender. Women over 60 years generally present with more extensive disease compared to men of the same age, and in the case of supra-inguinal involvement lesions are of greater severity (TASC type C or D) [5,6]. Some authors suggest these differences in severity at presentation could be explained by frequent osteo-articular comorbidities that mask the typical symptoms of PAD in women [5]. A meta-analysis using the National Inpatient Sample (NIS) database concluded that women have similar in-hospital mortality compared to men, who are more likely to develop periprocedural complications such as acute renal failure, gangrene or infections [7]. Another study published in 2017 describing the mid-term results of endovascular therapy of more than 1000 subjects (45% women) did not identify significant differences related to the subjects’ gender regarding one-year patency of endoprostheses [8]. Variables such as sex, age, diabetes, hypertension, dyslipidemia, or cerebrovascular disease were not predictors for restenosis or occlusion in the short and medium term. Predictors of restenosis and/or occlusion in the medium term were, for both sexes, renal dysfunction, poor distal arterial bed, and lesion length. Given that women generally have smaller-caliber vessels and poorer distal bed quality compared to men, one might assume they are at greater risk of restenosis or re-occlusion. However, no significant differences between the two sexes were evident, suggesting that women may benefit equally from endovascular therapies to reduce PAD-related morbidity and mortality. Most other studies in this area also failed to demonstrate a significant difference between genders in terms of patency, amputation, or mortality after percutaneous procedures. The objective of the present research is to identify potential differences between men and women undergoing peripheral angioplasty, as well as factors responsible for those differences.

## 2. Materials and Methods

In this prospective observational study, we included a total of 104 consecutive subjects (50 women and 54 men) who underwent PTA between January and October 2019 for symptomatic PAD in the Interventional Cardiology Laboratory of the Cluj-Napoca Rehabilitation Hospital, a high-volume center with two independent operators with more than ten years of experience each. Inclusion criteria were age over 18 years, debilitating intermittent claudication despite optimal medical treatment, or chronic limb-threatening ischemia. All subjects underwent a PTA procedure with either plain old balloon PTA and/or stenting. We excluded subjects who required amputation or surgical revascularization (aorto-iliac lesions > 5 cm and femoral–popliteal > 25 cm), or who were deemed unfit for revascularization and remained on conservative treatment.

Informed consent was obtained from all subjects and the following data were recorded: demographics, PAD history, presence of traditional cardiovascular risk factors, comorbidities, and the presence of associated coronary or cerebrovascular diseases. The values of the main biological parameters at admission, imaging data related to PAD, medical treatment prescribed, and procedural and anatomical characteristics were also recorded. The subjects were followed up clinically and using imaging for up to 2 years after the index revascularization, documenting clinical outcome and major adverse limb events (MALE). We defined MALE as severe limb-threatening ischemia requiring reintervention or major amputation. Death during follow-up was also recorded, although it was not considered an endpoint included in MALE.

Statistical analysis was performed using SPSS software version 21 (SPSS Inc., Chicago IL, USA). Descriptive statistics were used to summarize patient characteristics. Means and standard deviations were used for normally distributed continuous variables and medians and interquartile ranges = IQR for non-normally distributed continuous variables. Data are presented as mean ± standard deviation (SD). Continuous variables were compared using the Mann–Whitney U test and Student’s *t* test. Categorial variables were compared using the chi-square test and Fisher’s exact test. To ensure that we did not exclude any additional possible confounders, we performed 2 separate correlation matrixes to identify possible correlations between variables and MALE. Initial covariates were obtained from univariate analysis. For multivariate analysis adjusted odds ratio (OR) or hazard ratio (95% confidence interval [CI]) was used. Logistic regression with backward stepwise elimination was used to obtain patients’ characteristics in the final model, which were significantly associated with MALE. Kaplan–Meier survival analysis and log-rank test for comparison were performed to identify event-free limb-related adverse events. PAD class, smoking, age, and the presence of hypertension were identified as confounders by unstandardized beta-coefficients using multiple linear regression and were not used in the MALE prediction model. Additionally, we conducted a correlation matrix before performing multivariate analysis and found that MALE had a significant correlation with ABI and high triglyceride levels, and therefore was not included in the prediction model. A “*p*” value of 0.05 or less was considered statistically significant.

## 3. Results

The baseline characteristics of the subjects are summarized in Table 1.

The mean age of the subjects was 67 ± 10 years. Women were, on average, 4 years older than men (69 ± 10 years vs. 65 ± 9.2 years—*p* = 0.04). Concerning the major risk factors for atherosclerotic disease, statistically significant differences were detected only for smoking (almost 80% of male patients with PAD had a history of smoking, compared to just 44% of female subjects, *p* = 0.0004). No differences were found in the prevalence of arterial hypertension (82% vs. 83.3%, *p* = 0.93), dyslipidemia (70% vs. 65%, *p* = 0.72), obesity (20% vs. 17%, *p* = 0.88), or diabetes (46% for both) between the two groups.

Concerning the severity of PAD, the principal clinical criterion—the ankle–brachial index (ABI)—had a similar value in both groups, which was around 0.5 (*p* = 0.94). However, differences in Leriche–Fontaine class at presentation were also noted: the majority of female subjects were in stages III (18%) and IV (38%), respectively, compared to men, whose initial stage was predominantly IIB (57%—*p* = 0.035). A history of prior percutaneous or surgical revascularization did not differ significantly according to gender, the proportions being comparable between the two groups (12% vs. 18.5%; *p* = 0.51). Women had exclusively infra-inguinal involvement more frequently (78%), while men exhibited mixed disease, with supra + infrainguinal (37%) or solely supra-inguinal (20.3%) involvement (*p* = 0.0012). Additionally, the type of the lesion was different among genders, with women exhibiting more often complete occlusions (46 vs. 24%- *p* = 0.013) whilst combined lesions—stenosis + occlusion were more prevalent in the male group (16 vs. 40.8%—*p* = 0.009). In terms of procedural success and intra-operative complications, which might affect the overall outcome, there were no significant differences between the two groups.

The percutaneous therapy employed was also gender specific: women benefited mostly from simple balloon angioplasty (75.5%), while males were treated equally with balloon angioplasty and/or stenting (27 patients for each type of PTA—*p* = 0.03).

The mean follow-up of the two groups was 21 ± 2.4 months, with no significant differences between men and women. Two subjects in the male group were lost to follow-up by failing to complete any of the scheduled visits, while one subject in the female group was lost to follow-up after the first visit at 1-month post-index revascularization. There was a total of seven deaths during follow-up: three among women and four among men, none directly attributable to peripheral arterial disease (two strokes, three terminal neoplasms, and two of unknown cause). There were three major amputations in both groups, while surgical revascularization for recurrent limb-threatening ischemia occurred for five patients in the female group and only two in the male group. One patient who underwent dual-stent angioplasty for bilateral supra-inguinal involvement required emergency surgical revascularization for in-stent thrombosis 2 months post-PTA. The occurrence rate of MALE endpoints was similar in the two groups, at around 20%, as illustrated in Table 2. The Log Rank test applied to evaluate the distribution of survival according to sex did not return a statistically significant value (*p* = 0.914)—Table 3.

Figure 1 illustrates the Kaplan–Meyer curves indicating the prevalence of limb-related adverse events as a time function distribution in the two groups. As shown, there is no statistically significant difference in the incidence of MALE at 2 years between women and men. Therefor gender did not influence the incidence of PAD-related adverse events in our study.

The only parameter that proved to have a significant influence on the occurrence of limb-related adverse events (severe ischemia requiring percutaneous or surgical revascularization and major amputation) was ABI. An ankle–brachial index value below 0.5 was found to be an independent predictor of MALE (*p* = 0.001).

Table 4 and Figure 2 show the results of the Log Rank test and the Kaplan–Meyer curve for the distribution of the occurrence of adverse events according to the ABI value at enrolment.

No other analyzed variable was found to be an independent predictor for the occurrence of MALE except ABI. The threshold value from which the peripheral perfusion index becomes predictive for an unfavorable outcome is an ABI < 0.5.

## 4. Discussion

This study reveals gender-related anatomical and clinical characteristics of adverse limb outcomes in patients with PAD after percutaneous revascularization. At the time of enrolment, women had a tendency towards a more severe diseases compared to men. Consequently, the majority of male subjects had stable PAD with intermittent claudication as the main symptom (stages IIA and IIB Leriche–Fontaine, respectively). Meanwhile, 56% of women had limb-threatening ischemia, characterized by pain at rest and/or the presence of trophic lesions. According to a prospective multi-center study conducted by Riess et al. in Germany, men presented more commonly with intermittent claudication and ulcers or gangrene, whereas rest pain was predominant in women. In addition, the length of hospitalization was prolonged for female patients, who were more frequently referred to specialized rehabilitation facilities [9].

The anatomical site of the lesions was another feature peculiar to the female cohort: the vast majority of women (78%) had exclusively infra-inguinal involvement, compared to the male group where the distribution between supra, infra-inguinal, or mixt disease was much more homogeneous. However, the incidence of diabetes was comparable across the two groups, as it is known that diabetes is associated with a propensity for distal arterial bed disease. A similar observation by a group of Italian researchers led by Ferranti shows that women are more likely to have femoral–popliteal lesions compared to men, which negatively impacted the rate of adverse events after percutaneous revascularization, despite comparable overall survival rates in both sexes [10]. Furthermore, an influence of anatomical location on the therapeutic decision can be assumed: female subjects mainly benefited from balloon angioplasty, and only a very small proportion had stent implantation (24.5%). However, men were treated with simple balloon angioplasty and/or stent implantation in equal proportions. We can hypothesize that the predominance of infra-inguinal involvement in females, where the arterial diameter is smaller, justifies the use of balloon angioplasty as opposed to stent implantation. In men, on the other hand, the proportion of iliac lesions requiring revascularization with stenting per primam, as recommended by current guidelines, was higher [11]. Women are more likely to have overall poorer outcomes after peripheral vascular treatments due to their anthropometric and demographic factors, according to more recent data on mid- and long-term outcomes after vascular procedures [12].

Medium- and long-term outcomes of percutaneous and surgical revascularization in women are inconsistently reported in medical literature. The effect of gender on long-term revascularization outcomes for PAD is not well understood. Small, single-center studies on the short- and long-term effects of surgical vascular procedures in men and women have produced contradictory findings. The same is true for the outcomes of endovascular procedures, for which available literature contains limited and contradictory data. In some of these studies, women appeared to have equivalent outcomes to men, while in others, mid- and long-term outcomes were poorer, with higher rates of complications. Although the increased risk of adverse events in these studies may be attributed to a higher prevalence of diabetes, some authors have identified female gender as an independent predictor of complications [13]. A study by Egorova et al. indicated that mortality rates for women were persistently higher after amputations or surgical or endovascular procedures compared to men, but mortality and amputation rates improved over time. [14]. Similarly, women with critical lower limb ischemia were at increased risk of femoral–popliteal and infra-popliteal involvement. Despite similar success rates in post-PTA limb salvage, women with critical ischemia had an increased rate of subsequent adverse cardiovascular events, as shown in the study by McCoach et al. [15]. Furthermore, similar results were recorded in Ferranti’s research [9].

In contrast, several studies comparing the outcomes of peripheral percutaneous treatments in men and women did not identify the female gender as a predictor of unfavorable post-PTA course [16,17]. As for supra-inguinal angioplasty with initial stenting, the patency rate is dramatically reduced, and further treatments are frequently required to achieve sufficient clinical improvement and limb salvage [10]. Female gender was also associated with a higher prevalence of chronic limb-threatening ischemia, poorer patency, and more frequent restenosis for TASC C and D lesions, respectively, compared to men. Endovascular treatment of femoral–popliteal lesions provides equal results for both genders in terms of long-term patency [18]. A recent study by Choi et al. showed on a large group of patients with PAD undergoing PTA, and revealed that women had higher rates of death, myocardial infarction, or major amputation compared with men. Female subjects more frequently had complex arterial lesions, procedural complications, and MALE [19].

In the present study, there were no significant differences between sexes in the occurrence of major limb-related adverse events over a mean follow-up period of almost 2 years. The rates of major amputation and the need for reintervention were similar across the two groups, but, as indicated previously, the location of lesions and type of angioplasty in women were somewhat different. Although our research lacked the power to properly support a negative correlation due to its relatively small sample size, our findings are in line with other larger studies, which also failed to show a correlation between gender and MALE over time [8]. Orr et al. investigated 104 angioplasties, and there was no significant difference in the outcomes between men and women [17]. The composite risk of major adverse cardiovascular events between men and women was also similar, according to research conducted by Hussain et al. on 6915 patients [16]. The largest study on 25.658 patients published by Doshi et al. found no gender differences for in-hospital mortality after endovascular interventions [7].

The only independent predictor for the occurrence of MALE, regardless of gender, was the ankle–brachial index. A cut-off value below 0.5 was a strong negative predictor for MALE. This value is consistent with available scientific data, which indicate a value of 0.5 as the cut-off between chronic limb-threatening ischemia and clinically stable PAD [20]. In other older studies, an increased ten-year cardiovascular mortality rate of 37% was recorded for subjects with an ABI < 0.5, compared to a 27% mortality rate for patients with ABI values ranging between 0.5 and 0.7 and 22% in patients with ABI between 0.7 and 0.9 (*p* = 0.0039) [21]. Relative risk for all-cause mortality was higher (1.95, 95% CI 1.42–2.68) in subjects with ABI < 0.5 compared to subjects with ABI values ranging between 0.51–0.7 (RR 1.59, 95% CI 1.18–2.15). On the other hand, a systolic peripheral pressure of less than 50 mmHg is a better predictor of negative limb-related events [22].

To our knowledge, this is the first study that correlates an ABI value below 0.5 with the occurrence of MALE in a population with symptomatic PAD followed over an extended period of time after percutaneous revascularization. Historically, an ABI < 0.9 was significantly associated with higher coronary mortality, all-cause mortality, or myocardial infarction in both women and men. A pathological ankle–brachial index was an independent predictor of stroke in women only, while an ABI between 0.9 and 1 was statistically significantly associated with coronary death in both women and men [23]. Further studies are needed to establish a clear relationship between ABI value at presentation and MALE occurrence rate.

## 5. Conclusions

Even though the location of the atherosclerotic lesions, the Leriche–Fontaine stage at presentation, and the type of percutaneous treatment (simple balloon PTA vs. stent implantation) differed between the two sexes, there was no significant difference in the incidence rates of MALE over a 2-year follow-up period. Regardless of gender, an ankle–brachial index value below 0.5 was the single independent predictor of limb-related adverse events.

## Figures and Tables

**Figure 1 medicina-59-00480-f001:**
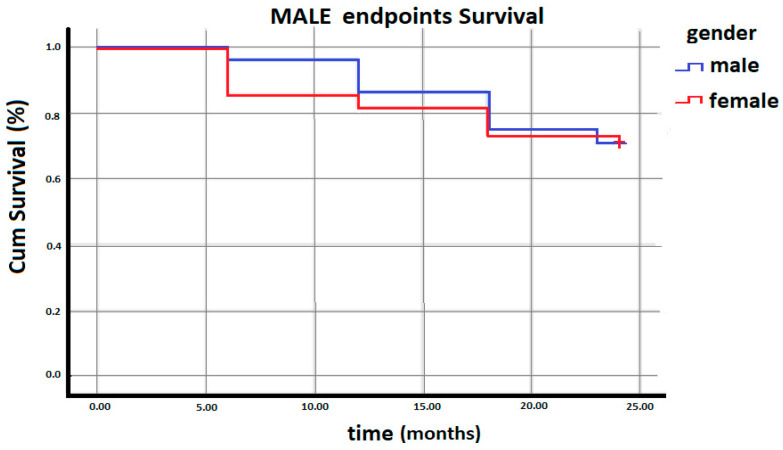
Kaplan–Meyer curves for MALE occurrence during follow-up.

**Figure 2 medicina-59-00480-f002:**
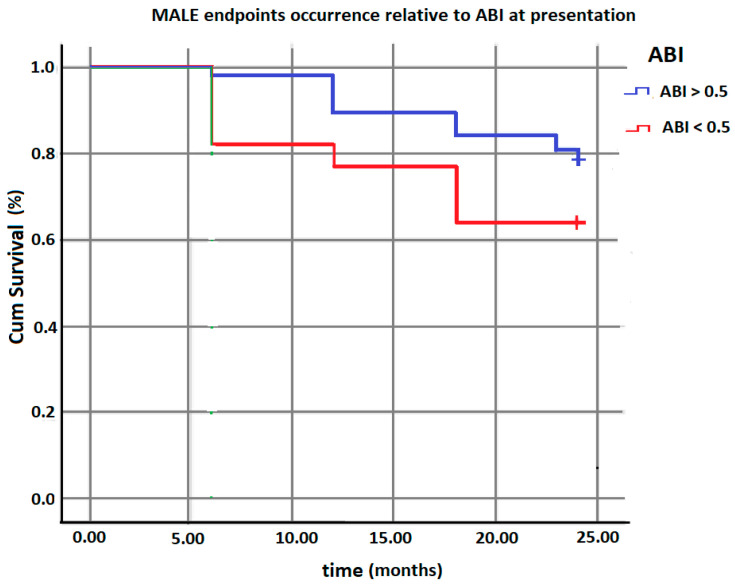
Influence of ABI at presentation on the occurrence of MALE in the two groups.

**Table 1 medicina-59-00480-t001:** Baseline characteristics of the two groups.

Parameter		Total *n* = 104	Female *n* = 50	Male *n* = 54	*p* Value (Men vs. Women)
Age mean ± SD		66.99 ± 10.2	69.04 ± 10.9	65.09 ± 9.2	*p* = 0.04 *
ABI Mean ± SD		0.51 ± 0.15	0.50 ± 0.16	0.51 ± 0.14	*p* = 0.94
Rhythm No (%)	AFib	12 (11.5)	7 (14)	5 (9.25)	*p* = 0.48
	Aflutter	1 (1)	0 (0)	1 (1.85)	
	SR	91 (87.5)	43(86)	48 (88.88)	
PAD class No (%)	IIA	2 (1.9)	0 (0)	2 (3.7)	*p* = 0.035 *
	IIB	53 (51)	22 (44)	31 (57.40)	
	III	22 (21.2)	9 (18)	13 (24.07)	
	IV	27 (26)	19 (38)	8 (14.81)	
Smoking No (%)	Yes	65 (62.5)	22(44)	43 (79.62)	*p* = 0.0004 *
	No	39 (37.5)	28 (56)	11 (20.37)	
Hypertension No (%)	Yes	86 (82.7)	41 (82)	45 (83.33)	*p* = 0.93
	No	18 (17.3)	9 (18)	9 (16.66)	
Obesity No (%)	Yes	19 (18.4)	10 (20)	9 (16.98)	*p* = 0.88
	No	84 (81.6)	40 (80)	44 (83.01)	
Dyslipidemia No (%)	Yes	70 (67.3)	35 (70)	35 (64.81)	*p* = 0.72
	No	34 (32.7)	15 (30)	19 (35.18)	
Diabetes No (%)	Yes	48 (46.2%)	23 (46)	25 (46.29)	*p* = 0.86
	No	56 (53.8%)	27 (54)	29 (53.7)	
Creatinine (mg/dL) Mean ± SD		0.96 ± 0.60 (0.85)	0.95 ± 0.82 (0.79)	0.97 ± 0.27 (0.89)	*p* = 0.84
LDL-cholesterol (mg/dL) Mean ± SD		111.93 ± 40.59	112.44 ± 36.68	111.46 ± 44.24	*p* = 0.87
PTA/bypass No (%)	Yes	16 (15.4%)	6 (12)	10 (18.51)	*p* = 0.51
	No	88 (84.6%)	44 (88)	44 (81.48)	
Unilateral or bilateral No (%)	Bilateral	43 (41.3%)	17 (34)	26 (48.14)	*p* = 0.2
	Unilateral	61 (58.7%)	33 (66)	28 (51.85)	
Supra or infrainguinal No (%)	Infra-inguinal	62 (59.6%)	39 (78)	23 (42.59)	*p* = 0.0012 *
	Supra + infra-inguinal	27 (26.0%)	7 (14)	20 (37.03)	
	Supra-inguinal	15 (14.4%)	4 (8)	11 (20.37)	
Type of PTA No (%)	Balloon	64 (62.1%)	37 (75.51)	27 (50)	*p* = 0.013 *
	Stent	39 (37.9%)	12 (24.48)	27 (50)	
Anatomical location of lesions No (%)	Infra-popliteal	12 (9.16)	6 (10.5)	6 (8.1)	*p* = 1
	Superficial femoral artery	68 (51.9)	28 (49.1)	40 (54.1)	*p* = 0.102
	Common femoral artery	6 (4.6)	2 (3.5)	4 (5.4)	*p*= 0.68
	Popliteal artery	19 (14.5)	12 (21.1)	7 (9.45)	*p* = 0.118
	Common iliac artery	11 (8.4)	4 (7)	7 (9.45)	*p* = 0.532
	External iliac artery	15 (11.44)	5 (8.8)	10 (13.5)	*p* = 0.274
Type of lesion No (%)	Occlusion	36 (34.6)	23 (46)	13 (24)	*p* = 0.013 *
	Stenosis	38 (36.5)	19 (38)	19 (35.2)	*p* = 0.839
	Combined	30 (28.9)	8 (16)	22 (40.8)	*p* = 0.009 *
Procedural success No (%)		91 (87.5)	43 (86)	48 (88.9)	*p* = 0.99
Flow-limiting dissections No (%)		3 (2.9)	2 (4)	1 (1.9)	*p* = 0.604
Significant residual stenosis No (%)		1 (1)	1 (2)	0 (0)	*p* = 0.476
Distal embolization No (%)		9 (8.65)	4 (8)	5 (9.2)	*p* = 0.99
Extensive lesion calcification No (%)		77 (74)	35 (70)	42 (77.8)	*p* = 0.512

* statistically significant.

**Table 2 medicina-59-00480-t002:** The mean survival estimation of the MALE endpoints between genders.

Mean * Survival Time (Months)—95% CI				
Sex	Estimated Survival	Std Error	Lower Bound	Upper Bound
Men	21.423	0.677	20.095	22.751
Women	20.449	0.966	18.555	22.343
Overall	20.95	0.583	19.808	22.093

* Estimation is limited to the largest survival time if it is censored.

**Table 3 medicina-59-00480-t003:** Log Rank (Mantel–Cox) test for distribution of survival curve between genders.

	h-Square	Difference	*p* Value
Log Rank (Mantel–Cox)	0.012	1	0.914

**Table 4 medicina-59-00480-t004:** Log Rank test (Mantel–Cox method) for the influence of ABI on MALE.

Overall Gender Comparison			
	h-Square	Difference	*p* Value
Log Rank (Mantel–Cox)	13.05	2	0.001 *

* statistically significant.

## Data Availability

Data is available upon reasonable request from the corresponding author.

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
