# Peer review of "Predictors for the Development of Major Adverse Limb Events after Percutaneous Revascularization—Gender-Related Characteristics"

_medicina, 2023, doi:10.3390/medicina59030480_

Round 1
Reviewer 1 Report
Comsa Horatiu and colleagues conducted a clinical study to assess the effect of gender on an occurrence of major adverse limb events (MALE) after PTA for symptomatic PAD. No such effect was found by the authors based on comparing Kaplan–Meier curves with the log rank test. Then the authors regrouped the subjects based on baseline ABI index value more or less than 0.5 and found the difference between these groups in MALE indicating statistically significant effect of ABI on the PTA outcomes. Also, authors concluded that some other factors (not clarified in the paper) have no significance in this regard.
Comments
1. The analysis of the data is not sufficiently described. For example, the Method section contains the statement, “Multivariate analysis was performed using the stepwise method” without any indication what kind of the analysis was performed, what variables were included in the model(s), what was the goodness-of-fit of the model(s), what was the stepwise approach - forward, backward, how significance of the relationships was evaluated, etc. The comparison of baseline data appears to be done using an unpaired t-test, however, currently it is a matter of guess for readers, preventing their assessment of the validity of the conclusions based on this comparison. The text, “Roc, Kaplan-Meier, and Log rank curves were utilized” is quite confusive, at least. Roc probably means ROC analysis whereas Log rank is not a curve at all, it is the statistical test.
2. No proper lesion characteristics are given. The relation to Poupart ligament is not sufficient. Lesion length, diameter, presence of calcification greatly affect the results of the PTA.
3. Also, the procedural aspects are necessary to properly interpret the outcome data. Technical success, procedural complications such as distal embolization, residual stenosis, flow-limiting dissections, operator experience are all important for predicting the outcomes.
4. How the ABI with the threshold of 0.5 was chosen for respective group analysis?
5. The study findings are largely negative. What gives the authors confidence to conclude that the insufficient significance of the relationships is an evidence of the lack of these relationships and not the result of insufficient sample size? The proper power analysis is needed to support the negative claims.
6. The p for between-group comparison of Creatinine is extremal error - the difference between means: 0.97-0.95=0.02 the standard error of the difference with the raw deviations 0.82 and 0.27 is 0.164, then t for group sizes 50/54 should be around 0.1 whereas p - around 0.9, but by no means 0.0043.
7. The illustrations should bear proper labeling for sex, e.g. men/women or male/female but not 1.00/2.00
Author Response
Reviewer 1 comments
- Thank you for pointing this out to us! The Methods section of the manuscript was reformulated as to better describe the statistical analysis of the data. We corrected the typographical error. Statistical analysis was performed using SPSS software version 21 (SPSS Inc., Chicago IL, USA). Descriptive statistics were used to summarize patient characteristics. We used means and standard deviations for normally distributed continuous variables and medians and interquartile ranges=IQR for nonnormally distributed continuous variables. Data are presented as mean ± standard deviation (SD). Continuous variables were compared using the Mann-Whitney U test and the Student t test. Categorial variables were compared using the chi-square test and Fisher’s exact test. To ensure that we did not exclude any additional possible confounders we performed 2 separate correlation matrixes to identify possible correlations between variables and MALE. Initial covariates were obtained from univariate analysis. For multivariate analysis adjusted odds ratio (OR) or hazard ratio (95% confidence interval [CI]) was used. Logistic regression with backward stepwise eliminaton was used to obtain patients’characteristics in the final model which were significantly associated with MALE. Kaplan-Meier survival analyis and log-rank test for comparison were performed to identify event-free limb-related adverse events. A "p" value of 0.05 or less was considered statistically significant.
2-3. Table 1- Baseline characteristics of the subjects has been updated to include both anatomical and procedural aspects related to PTA in both groups. Also, statistical analysis was redone in order to include the new data. Center and operator experience was mentioned in the Methods section as it is indeed relevant to procedural outcome.
- We established the threshold of 0.5 for ABI based on multiple older studies and statements which have found this cut-off to be quite indicative of severe PAD. The manuscript was updated with references to those respective papers in the discussion section.
- We agree with the reviewer. Insufficient statistical significance between variables and MALE can be due to lack of association or insufficient sample size. In order to detect a 50% reduction in MALE (power 80%, type 1 error 5%), 600 patients had to be included in the trial. We acknowledge that the size of our sample is small, but other studies on comparable numbers of patients have indicated no differences in outcomes between men and women who underwent endovascular procedures: Orr et al. investigated 104 angioplasties and there was no significant difference in the outcomes between men and women The composite risk of major adverse cardiovascular events between men and women was also similar according to the research conducted by Hussain et al. on 6915 patients. The largest study on 25.658 patients published by Doshi et al, found no gender differences for in-hospital mortality after endovascular interventions.
- Creatinine ‘p’ was recalculated and results updated in table 1. The p value is 0.84
- We updated the ilustrations with proper and clear labeling.
Sincerely,
Dr. Gabriel Cismaru, MD, PhD
Department of Cardiology,
Clinical Rehabilitation Hospital,
University of Medicine and Pharmacy “Iuliu Hatieganu” Cluj-Napoca
46-50 Viilor street, Cluj-Napoca, 400347, Romania
Comsa Horatiu
Gusetu Gabriel
Rosu Radu
Dumitru Zdrenghea
David Adina
Dutu Bogdan
Tomoaia Raluca
Fringu Florina
Irimie Diana
Pop Dana

Reviewer 2 Report
I would like to thank the authors for there work
This is a well-written manuscript with interesting evidence and future perspective. Overall I have few recommendations:
1. The manuscript could be strengthened by adding more pathophysiology and clinical relevance to the introduction to clarify the importance of the conducted research and its clinical impact.
2. Did you consider the PAD class and smoking as confounders while doing your statistical analysis ? if so please explain (how) in the method .... They are both different in the male and female groups and they can affect your results.
3. Could you please clarify if the measured organ function tests are within the normal range or not and if creatinine clearance was also considered as a confounders in your statistical analysis
Minor recommendations:
1. In table 1 could you please add * to the significant p values to make them more clear for the readers
2. Could you please add the units for all the measured parameters
3. Please add the p value for LDL-cholesterol (you already reported other non-significant p values , only this one was reported as NS)
4. For all the p-values in table 1 ,could you please clarify that this the difference between the male and female groups, it is mentioned in the results section however not clear in the table as you reported 3 groups.
Author Response
Reviewer 2 comments
- The Introduction section was updated as to include more clinical, epidemiological and pathophysiological data in order to better underline the relevance of the present research.
- Yes, PAD class, smoking, age and the presence of hypertension were identified as confounders by unstandardized beta-coefficients using multiple linear regression. They disturb the relationship with MALE and could not be used in the prediction model.
Additionally, we conducted a Correlation matrix before performing multivariate analysis and found that MALE had a significant correlation with ABI and high trigliceride levels, therefor not included in the prediction model. (if needed we can provide the coefficient values, correlations and uni+multivariate analyses, but these are large tables that are of little interest to the average reader).
- The measured parameters of organ function were all within normal range. The absolute value of creatinine, which multiple linear regression did not identify as a confounder, was utilized instead of creatinine clearance.
Minor recommendations:
- Table 1 was revised and significant p values were marked with ‘*’
- Units were added for all parameters in Table 1.
- p value for LDL-cholesterol was added in Table 1.
- Made it clearer in Table 1 that p is a measure statistical comparison between the male and female group.
Sincerely,
Dr. Gabriel Cismaru, MD, PhD
Department of Cardiology,
Clinical Rehabilitation Hospital,
University of Medicine and Pharmacy “Iuliu Hatieganu” Cluj-Napoca
46-50 Viilor street, Cluj-Napoca, 400347, Romania
Comsa Horatiu
Gusetu Gabriel
Rosu Radu
Dumitru Zdrenghea
David Adina
Dutu Bogdan
Tomoaia Raluca
Fringu Florina
Irimie Diana
Pop Dana
